# Treating Neurodegenerative Disease with Antioxidants: Efficacy of the Bioactive Phenol Resveratrol and Mitochondrial-Targeted MitoQ and SkQ

**DOI:** 10.3390/antiox10040573

**Published:** 2021-04-08

**Authors:** Lindsey J. Shinn, Sarita Lagalwar

**Affiliations:** Skidmore College Neuroscience Program, Saratoga Springs, NY 12866, USA; lshinn@alumni.skidmore.edu

**Keywords:** antioxidants, resveratrol, MitoQ, SkQ, neurodegenerative disease

## Abstract

Growing evidence from neurodegenerative disease research supports an early pathogenic role for mitochondrial dysfunction in affected neurons that precedes morphological and functional deficits. The resulting oxidative stress and respiratory malfunction contribute to neuronal toxicity and may enhance the vulnerability of neurons to continued assault by aggregation-prone proteins. Consequently, targeting mitochondria with antioxidant therapy may be a non-invasive, inexpensive, and viable means of strengthening neuronal health and slowing disease progression, thereby extending quality of life. We review the preclinical and clinical findings available to date of the natural bioactive phenol resveratrol and two synthetic mitochondrial-targeted antioxidants, MitoQ and SkQ.

## 1. Introduction

Neurons most vulnerable to neurodegenerative disease tend to be large, highly innervated, highly branched, and highly plastic [1] including hippocampal and cortical pyramidal neurons in Alzheimer’s, striatal spiny neurons in Huntington’s, and cerebellar Purkinje neurons in the Spinocerebellar ataxias. The considerable reliance of these neurons on oxidative phosphorylation for ATP production, long-distance mitochondrial trafficking, and dynamic fusion and fission needs makes neurons uniquely vulnerable to mitochondrial dysfunction and oxidative stress [2,3].

The hint that mitochondrial dysfunction contributes to neurodegenerative disease was provided through genetics. Mutations in the superoxide dismutase gene cause familial Amyotrophic Lateral Sclerosis (ALS) and lead to an overproduction of reactive oxygen species (ROS) in motor neuron mitochondria; a feature seen in both familial and sporadic ALS [4,5], albeit due to different causes. Mutations in mitochondrial-localized PINK1 and cytosol-localized Parkin affect mitochondrial fidelity and cause recessive forms of Parkinson’s disease [6]. Additionally, dysfunction in complex I of the mitochondrial electron transport chain in dopaminergic neurons is associated with the development of Parkinson’s disease [7]. Despite the lack of known mitochondrial genetic involvement, decreased electron transport chain activity accompanies the development of Alzheimer’s disease [8], Huntington’s disease [9], and Spinocerebellar ataxia type 1 [10,11].

Antioxidants have proved efficacious in ameliorating oxidative stress and reducing neurodegenerative pathologies in cell culture [12,13,14] and invertebrate models of disease [15]. Here, we review the pre-clinical mammalian animal model studies and clinical trials of the antioxidants resveratrol, MitoQ, and SkQ against neurodegenerative disease phenotypes (summarized in Table 1). If ultimately successful, these small molecule compounds would provide non-specific, non-invasive, and accessible therapies that improve the quality of life by slowing disease progression in patients.

## 2. Resveratrol

Trans-resveratrol (3,5,4′-trihydroxy-trans-stilbene, referred to here as resveratrol) has been widely tested in multiple studies as a treatment for a variety of diseases and conditions such as cancer, heart disease, cholesterol, and age-related neurodegenerative diseases. It is a polyphenol, or a micronutrient, typically found in grapes to produce their red hue. It was first discovered in the 1940s when it was extracted from the roots of white hellebore (*Veratum grandiflorum O. Loes*), a poisonous plant that originates from Europe and Western Asia. It was not until 1992 that researchers observed its involvement in cardio-protective effects in red wine [16]. Following this discovery, resveratrol was examined through scientific studies for its ability to inhibit carcinogenesis. Although Timmers and colleagues [17] found some contrasting data regarding resveratrol’s supporting effects on the NAD-dependent deacetylase sirtuin-1 (SIRT1) in non-human subjects, overall, there is stronger evidence that resveratrol may inhibit symptoms of age-related diseases in animal models through an innate antioxidant effect.

### 2.1. Preclinical Studies

Sharma and Gupta [18] first investigated resveratrol treatment in a sporadic Alzheimer’s-type rat model. Cognitive impairment was initiated using an established method of bilateral intracerebroventricular injection of streptozotocin [19] on days 1 and 3 while undergoing learning and memory assessments during treatment [18]. Additionally, lipid oxidation was measured through malondialdehyde analysis, and oxidative stress was measured through glutathione analysis. Resveratrol (10 and 20 mg/kg i.p.), administered continuously for 21 days beginning from the first day of streptozotocin injection, had no effect on body mass or mortality. Resveratrol improved memory acquisition and retention in a dose-dependent manner in both the passive avoidance test and the elevated plus maze, without affecting spontaneous locomotor activity as measured by closed field activity. Malondialdehyde and glutathione levels in whole brain tissue were restored to those of the sham-injected rats with resveratrol treatment [18]. These findings support the beneficial properties of resveratrol to inhibit cognitive impairment and oxidative stress induced by streptozotocin.

In a larger study, Huang et al. (2011) [20] examined resveratrol on amyloid-beta (Aβ) induced hippocampal neuron loss and memory dysfunction in adult male Sprague-Dawley rats. The rats were injected with 0.5 μM/min of Aβ (100 μM/ 5 μL) directly into the lateral ventricle through a Hamilton microsyringe and mini pump. The injections lasted 10 min and the syringe was left on the injection site for 2 min to ensure Aβ infusion. Using the same procedure, Aβ was injected once per day for 7 days followed by resveratrol (100 μM/5 μL, i.c.v.) injection after 30 min. Hippocampal spatial memory was assessed by the Morris water maze (MWM) and motor deficits were assessed by the accelerating rotarod test. Following these procedures, the rats were decapitated and hippocampi were analyzed for Aβ load, iNOS, heme oxygenase-1, and lipid peroxidation [20]. Resveratrol significantly reduced hippocampal amyloid load, iNOS, and lipid peroxidation, while increasing heme oxygenase-1 levels compared to non-treated Aβ-injected animals. Histological examination of hippocampal CA1 and CA3 tissue showed that resveratrol diminished neuronal shrinkage and vacuole formation caused by Aβ. Finally, resveratrol improved hippocampal-based spatial memory deficiencies by decreasing escape latency in the Morris water maze. Neither Aβ injection nor resveratrol treatment had any effect on rotarod performance [20]. The study further demonstrated resveratrol can substantially reverse hippocampal oxidative stress, neuronal atrophy, and memory deficits triggered by Aβ.

To further examine resveratrol’s response to toxicity, Abolaji et al. (2018) [15] focused on the interaction between MPTP-induced neurotoxicity in *Drosophila melanogaster* and resveratrol. *Drosophila* was orally exposed to resveratrol (30 and 60 mg/kg) and MPTP (up to 120 mg/kg) for 3 days of treatment followed by assessment of cell viability and oxidative damage. Overall, resveratrol treatment reversed MPTP-induced behavioral deficits (increased climbing rates and fly emergence), cellular deficits (increased AChE activity and MTT-based cell viability), and oxidative damage (reduced peroxide, nitric oxide, and glutathione-S-transferase activity levels, increased catalase activity), while preventing MPTP-induced brain lesions. Resveratrol also affected a dose-dependent increase in fly longevity [15]. While this study focused on resveratrol’s influence on longevity, the findings strongly suggest that resveratrol holds therapeutic potential against neurodegenerative disease. 

Following resveratrol’s anti-aging potential, Corpas et al. (2019) [21] analyzed resveratrol’s influence on wild type and Alzheimer 3xTg-AD mice to determine resilience against neurodegeneration. Mice were orally administered 100 mg/kg resveratrol from age 2–10 months and were behaviorally assessed by an open field, dark and light, Morris water maze, Boissier’s four-hole board, and novel object recognition. Resveratrol improved exploratory behavior and reduced anxiety in both mouse strains. Cognitive performance was enhanced and hippocampal Aβ and p-tau pathology were reduced by resveratrol in the 3xTg-AD mice. Additionally, resveratrol promoted Aβ catabolism and upregulated the SIRT1 longevity pathway, specifically the activation of PGC-1∝ and CREB [21]. These findings are rather significant. Not only did resveratrol clear pathology and improve overall cognitive capabilities, but the long-term treatment also led to genomic and proteomic changes in vulnerable tissues that may provide long-term protective effects. 

### 2.2. Clinical Studies

Following a large body of research investigating resveratrol’s anti-aging and neuroprotective effects in animal models, clinical trials were implemented to test resveratrol’s efficacy in humans. Noting that resveratrol showed enhanced performance in overweight adults [17,22,23], Witte et al. (2014) [24] conducted a double-blind placebo-controlled intervention to observe whether resveratrol can increase memory performance in healthy overweight (BMI of 25–30 kg/m^2^) individuals, aged 50–80. Control participants who received a placebo (50–75 years old) were matched for sex, age, and BMI. Baseline cognitive, psychiatric, and neurological exams were administered prior to a daily intake of resveratrol for 26 weeks (200 mg). In an auditory verbal learning task, increased memory retention performance—defined as the number of correctly recalled words on the fifth learning trial subtracted from the number of correctly recalled words following a 30-min delay, delayed recall—defined as the number of correctly recalled words following the 30-min delay, and recognition—defined as the number of correctly recognized words out of a newer list of words, significantly improved following resveratrol, but not placebo, treatment. Learning ability—defined as the total number of correctly recalled words following each of the five learning trials, improved compared to baseline in both the resveratrol and placebo groups. fMRI assessment showed increased resting-state functional connectivity in anterior and posterior hippocampi following resveratrol treatment compared to placebo. No significant changes in hippocampal volume or total gray matter volume were observed. Diabetes and adiposity risk decreased with resveratrol treatment, but not with placebo treatment, compared to the baseline as measured by decreased glycated hemoglobin (HbA1c) and increased leptin [24]. Overall, the study supports resveratrol as a facilitator of caloric restriction that enhances glucose metabolism and subsequently leads to improved neuronal functional connectivity and memory performance. 

Building on resveratrol’s caloric restriction-mediated cognitive improvement, Turner et al. (2015) conducted a 52-week placebo-controlled double-blind trial of resveratrol treatment in patients with mild to moderate Alzheimer’s while evaluating biomarker and volumetric MRI as well as clinical results [25]. Participants (*n* = 119) were randomized to oral administration of placebo or resveratrol (500 mg) daily with a dose increase by 500 mg every 13 weeks, ending with 1000 mg twice daily. MRI imaging, CSF, and plasma were collected at baseline and following completion of treatment. Surprisingly, CSF Aβ40 and plasma Aβ40 levels decreased significantly over the course of the trial in placebo-treated but not in resveratrol-treated groups. Resveratrol seemed to have a stronger effect on CSF and plasma Aβ42 levels, although the decrease was not as pronounced as in the placebo-treated controls. Furthermore, brain volume and ventricular size increased with resveratrol treatment more prominently than with placebo treatment [25]. The authors note that their findings are not associated with cognitive decline, and the change in CSF Aβ40/42 suggests that resveratrol penetrates the blood-brain barrier. Importantly, high doses of resveratrol were safe and well-tolerated. 

The same research group next performed retrospective analysis on their previously banked CSF and plasma samples [25] by multiplex assay of neurodegenerative markers and metalloproteinases (MMPs) [26]. In addition to the amyloid changes described above, resveratrol treatment significantly reduced CSF levels of MMP9, an enzyme upregulated in several brain pathologies [27], and increased macrophage-derived chemokine (MDC), IL-4, and FGF-2. Additionally, resveratrol increased plasma MMP10 and decreased IL-12P40, IL12P70, and the T-cell activator RANTES. Taken together, the multiplex results indicate that long-term, high dosage treatment of resveratrol may reduce neuroinflammation and promote innate immunity in the brain.

## 3. MitoQ

MitoQ is a mitochondria-targeted antioxidant composed of ubiquinone and triphenylphosphonium (TPP+) [28] that has produced efficacious amelioration of mitochondrial ROS. The TPP+ moiety in MitoQ allows its buildup in the mitochondria, where it is absorbed into the inner membrane and eventually recycled into active quinol by Complex 2 of the respiratory chain. With its protective properties, many studies have examined MitoQ to combat diseases including Alzheimer’s, Parkinson’s, and Friedreich’s ataxia. However, since its discovery in the 1990s, there have yet to be many conclusive clinical trials. Despite this, multiple preclinical studies strongly promote the beneficial effects of MitoQ against neurodegenerative disease.

### 3.1. Preclinical Studies

An early study by Ghosh et al. (2010) [29] examined the effects of MitoQ in MPTP- mouse models of Parkinson’s disease. Six to 8 week old male C57BL/6 mice were given 4 mg/kg oral dosage of MitoQ for 1 day prior to MPTP administration (25 mg/kg i.p. once daily for 5 days), 5 days during MPTP administration, and 7 days following MPTP treatment. Control mice received saline. The study does not clearly identify the results of the three different treatments, and combines the results under the label “MitoQ-treatment”. What is clear, however, is that “MitoQ-treatment” increased MPTP-induced depletion of dopamine, the dopamine metabolite DOPAC, and homovanillic acid in striatal tissue comparable to control-treated levels; restored tyrosine hydroxylase expression levels in striatal and substantia nigral neurons of MPTP-treated mice; enhanced locomotor activity to control levels; and lowered the unpaired electron spin resonance spectra in striatum and substantia nigra [29]. 

Turning to mouse models of Alzheimer’s, McManus et al. (2011) [30] examined the protective effects of MitoQ against Alzheimer-like pathology and symptoms in female 3xTg-AD mice. At two months of age, transgenic and control wild type mice were either administered MitoQ (100 μM) or the negative control for MitoQ (dTPP; 100 μM) ad libitum in their drinking water for 5 months. MitoQ prevented the onset of cognitive deficits in the 3xTg-AD mouse as shown by significantly decreased escape latencies during Morris water maze trials comparable to wild type levels. Whole brain tissue analysis revealed MitoQ decreased oxidative stress and lipid peroxidation (measured by prevention of decreased GSH/GSSG ratios and malondialdehyde levels), astrogliosis, and caspase-3/7 activity. Additionally, Aβ42 immunoreactivity in the hippocampus and cortex was reduced, further supporting its therapeutic effects in neurodegenerative disease [30]. 

Stucki et al. (2016) aimed to determine the cellular factors that provoke mitochondrial dysfunction in Spinocerebellar ataxia type 1 (SCA1) progression in Sca1^154Q/2Q^ mice at a symptomatic stage while observing MitoQ’s restorative properties against SCA1 pathology [10]. MitoQ (MS010; 500μM) was administered to wild type (WT) and Sca1^154Q/2Q^ mice ad libitum through the drinking water twice a week for either 3 weeks or 16 weeks, while control mice were given regular drinking water. In particular, the long-term treatment improved mitochondrial morphology and electron transport chain activity in Sca1^154Q/2Q^ cerebellar Purkinje cells, increased Purkinje cell number, and restored the motor phenotype of Sca1^154Q/2Q^ mice by rotarod and hind-limb clasping assessment [10]. 

Taken together, the MitoQ pre-clinical findings summarized above indicate the non-specific therapeutic potential of MitoQ-treatment across multiple neurodegenerative diseases involving multiple regional targets.

### 3.2. Clinical Studies

MitoQ has been assessed in multiple clinical trials including one of Parkinson’s patients [31,32,33] and appears to be safe at varying doses. However, an improvement of neurodegenerative symptoms by MitoQ is as of yet unseen [32].

## 4. SKQ1 and SKQR1

Like MitoQ, SkQ1 and SkQR1 fall under the SkQ class of mitochondrially-targeted antioxidants. In general, SkQ molecules contain an antioxidant moiety linked to a lipophilic cation, allowing for transport into the negatively charged mitochondrial membrane, by a hydrocarbon chain, to facilitate membrane transport [34]. While not as widely tested as MitoQ, SkQ1, in particular, has begun to show efficacy in mice models and is currently undergoing clinical trials for dry eye and macular degeneration [35,36].

### 4.1. Preclinical Studies

An early (2014) study by Stelmashook, et al., treated wild type Wistar rats with SkQ1 to deduce whether SkQ1 might affect memory or long-term potentiation [37]. A single i.p. injection of SkQ1 (1 μmol per kg of body weight) was given at the age of 3–4 months. One day after injection, the animals were tested for passive avoidance conditional reflex. Following a painful stimulus in the dark chamber, SkQ1-injected rats spent significantly greater time avoiding the dark chamber than saline-treated control rats. SkQ1 was injected into 1-month old rats 24 or 48 h prior to hippocampal slice preparation. In the 24-h slice preps, but not the 48-h preps, long-term post-tetanic potentiation was maintained over a 75-min recording period following high-frequency stimulation from mice.

Following that initial work in wild type rats, multiple studies have assessed the protective effects of SkQ1 on neurodegenerative pathologies in the senescence-accelerated OXYS rat, a model of aging. The OXYS rat is an inbred strain featuring an overproduction of free radicals, lipid peroxidation, protein oxidation, and DNA damage, resulting in a variety of neurodegenerative features [38]. 

Loshchenova and colleagues [39] compared the amount of mitochondrial DNA and the prevalence of deletion mutations in mitochondrial DNA in the hippocampus of OXYS rats and wild type Wistar rats at postnatal and older ages. Regardless of genotype, mitochondrial DNA quantity increased during the postnatal period prior to a decline in young adulthood that persisted through older age. Concomitant progression of neurodegenerative features in the OXYS rats did not affect mitochondrial DNA quantity. In contrast, mitochondrial DNA deletions accumulated in OXYS rats to a maximum high of 2.3 times as much as in Wistar rats at 10 days of age. As the animals aged, the rate of accumulation decreased; however, it remained statistically higher in OXYS rats compared to Wistar rats through 6 months of age. Administration via dietary supplementation of SkQ1 (250 nmol per kg of body weight) from the ages of 1.5 months to 3 months of age decreased mitochondrial DNA deletions in both Wistar and OXYS rats. Moreover, SkQ1 administration decreased neurodegenerative-associated “passive behaviors” in OXYS rats as determined by the number of square crossings and rearings in an open field test [39]. 

In a larger study aimed at assessing the effects of treatment during neurodegenerative disease progression, Stefanova, et al. (2016) [40] administered SkQ1 (250 nmol per kg of body weight) in OXYS and Wistar rats from age 12 months through 18 months. Mitochondrial improvements in the hippocampal CA1 neurons of treated OXYS animals included significant increases in mitochondrial area, increased expression of mitochondrial fission and fusion genes Drp1 and Mfn2, respectively, and increased enzymatic activity of electron transport chain complexes I and IV. Particularly significant, complex IV activity increased by 30%. Hippocampal neuronal integrity improved with treatment as well. Specifically, treatment in OXYS rats reduced neuronal loss of CA1, CA3, and dentate gyrus neurons; increased soma volume in CA3 and dentate gyrus neurons; and increased nuclear volume in CA1 neurons. Ultrastructural characterizations of CA1 neurons showed that SkQ1 treatment reduced mitochondrial swelling and lipofuscin accumulations. 

Interestingly, Stefanova and colleagues assessed the effect of SkQ1-treatment of OXYS rats on BDNF growth factor and its receptors in the CA1 [40]. Treatment increased mature BDNF levels and decreased pro-BDNF levels while keeping total BDNF constant. Treatment also elevated the ratio of the phospho-TrkB/TrkB receptor ratio, and reduced expression and pro-BDNF co-localization of the p75^NTR^ receptor. The authors attributed these changes to SkQ1 promotion of neurogenesis and cell survival. A supportive factor to this theory was their finding that in the CA1, SkQ1 treatment of OXYS rats increased the number of excitatory asymmetric synapses by 50% and decreased the number of inhibitory symmetric synapses by 60%. Ultrastructural characterization showed that treatment reduced synaptic vacuole formation, swelling, and disruption of mitochondrial cristae. Additionally, treatment increased the number of active zones, as determined by the presence of neurotransmitter vesicles, pre-synaptic synapsin I expression, and postsynaptic PSD-95 expression.

To address the effects of SkQ1 treatment on Alzheimer’s-type pathology and behavior in the OXYS rat, Stefanova et al. confirmed by biochemical assay and immunostaining measures that 18-month old OXYS rats feature elevated hippocampal beta amyloid Aβ_1-40_ and Aβ_1-42_ [40]. SkQ1 treatment reduced both species of beta amyloid. Tau protein levels and tau phosphorylation was measured by Western blotting. Compared to Wistar rats, OXYS hippocampal tau is elevated, and phosphorylation of its threonine 181, serine 262, and serine 396 (the latter being phosphorylated in paired helical filament tau) are elevated as well. SkQ1 treatment does not reduce levels of S396-phoshorylated tau, but does reduce T181- and S262-phosphorylated tau levels along with total tau levels. Hippocampal-dependent learning and memory were assessed by a Morris water maze. SkQ1 treatment decreased escape latencies of OXYS rats significantly on all five trial days. By the 6th testing day, the time spent in the target quadrant by treated rats was elevated by over 50% compared to untreated OXYS rats and was indistinguishable from Wistar rats. 

Expanding on the Stefanova et al. study, Kolosova, et al. (2017) [41] administered SkQ1 (250 nmol per kg of body weight) during a period of severe stage neurodegenerative disease in OXYS rats, age 19–24 months. At this age, OXYS rats display reduced physical activity, neurological deficits such as increased foci of demyelination and increased ventricular volume, augmented production of serum and hippocampal Aβ_1-40_ and Aβ_1-42_, and numerous hippocampal mitochondrial deficits. 

SkQ1 treatment increased the number of squares crossed in an open field test in both Wistar and OXYS rats, as well as increased the frequency of rearings in OXYS rats. Hippocampal, but not serum, Aβ_1-40,_ and Aβ_1-42_ levels were significantly decreased with treatment in both genotypes. With respect to mitochondrial morphology, SkQ1 treatment reduced the number of mitochondria, increased the average area of mitochondria, and increased the number of mitochondrial-endoplasmic reticulum (MAM) contacts in the CA1 region of the OXYS hippocampus. Notably, for the older age rats used in this study, SkQ1 treatment was not able to reverse or slow ventricular enlargement or demyelination [41].

In addition to the amelioration of Alzheimer’s/aging characteristics in OXYS rats, the effects of SkQ1 treatment were assessed in an MPTP model of Parkinson’s disease in C57BL/6 mice [42]. Pavshintsev et al. (2017) administered SkQ1 (1000 nmol per kg) to 8-week old mice by i.p. injection daily for one week in control intact mice and in “Parkinson’s” mice three days following a four-day i.p. injection of MPTP. Therefore, treatment occurred after animals had reached a Parkinson-like state. In striatal tissues, SkQ1 injection increased tyrosine hydroxylase and dopamine levels in MPTP mice, although there was no effect on the dopamine precursor DOPA or metabolite DOPAC. Immunohistochemical staining revealed increased tyrosine hydroxylase expression and increased nuclear counts of tyrosine hydroxylase-positive neurons in the substantia nigra and ventral tegmental area. The authors also assessed the effects of treatment on behavior. Motor ability, as determined with distance walked in the open field paradigm, and sensorimotor ability, as determined with the beam walking test, were both improved in MPTP mice following SkQ1 injection. The “Porsolt forced swim” was used to test for affective side effects from the MPTP or SkQ1 injections, and no differences among groups were found.

### 4.2. Clinical Studies

Although just outside the scope of this review, SkQ1 showed promising results in treating retinopathy in the OXYS rats [43] and in a light-induced retinopathic albino rat model [44]. It also improved myelin-depletion in an LPS-induced oligodendrocyte cell culture model of multiple sclerosis [45]. SkQ1 is currently undergoing clinical trials by the company Mitotech, where it is in Phase II trials for a macular degeneration study, in Phase III trials for a dry eye treatment study, and has completed the preclinical phase of a multiple sclerosis study.

## 5. Conclusions

Preclinical and clinical studies overwhelmingly show resveratrol, MitoQ, and SkQ treatments to be safe at chronic, high dosages. However, discrepancies exist in the efficacy results of animal model studies compared to human trials. There are various reasons for this. Dysfunctional mitochondria may not provide abundant antioxidant substrates. Similarly, mitochondrial-targeted antioxidants, such as MitoQ and SkQ, may not be as efficacious in damaged mitochondria with deficient electron transport chains. Additionally, aged human neurons undergoing degeneration may not take in therapeutics as readily as less vulnerable rodent neurons. Finally, interference by glial cells during neurodegenerative disease-induced gliosis may affect therapeutic efficacy as well. Therefore, dosage, treatment length, disease stage, and mode of delivery in patient participants need further investigation, though the strength of the mouse model data provides strong support for further research. We also note the potential prophylactic benefits of antioxidants in healthy individuals who may be at risk for developing the disease.

## Figures and Tables

**Table 1 antioxidants-10-00573-t001:** Outcomes, by reference number, of resveratrol, MitoQ and SkQ1/SkQR1 treatment in pre-clinical studies.

	Pre-Clinical Outcomes	Effect of Treatment	Resveratrol	MitoQ	SkQ1/SkQR1
**Mitochondrial Integrity**	ETC complex activity	Increased		10	40
Mitochondrial content	Increased			40,41
Mitochondrial morphology	Restored		10	
mtDNA deletions	Decreased			39
Fusion/fission gene upregulation	Increased			40
Oxidative stress	Decreased	15,18	30	
**Cellular Integrity**	Neuronal function biomarkers	Increased	15,20	10,29,30	40,42
Lipid peroxidation	Decreased	18,20	30	
Neuronal atrophy	Decreased	15,20		40
Vacuole formation	Decreased	20		
SIRT1 pathway activation	Increased	21		
**Pathological Benchmarks**	Amyloid load	Decreased	20,21	30	40,41
Phosphorylated tau	Decreased			40
Astrogliosis	Decreased		30	
**Behavioral Benchmarks**	Spatial memory	Increased	15,20	30	40
Locomotor activity	Increased		29,30	42
Passive behaviors (open field test)	Decreased			39,41
Longevity	Increased	15

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
