# Peer review of "Treating Neurodegenerative Disease with Antioxidants: Efficacy of the Bioactive Phenol Resveratrol and Mitochondrial-Targeted MitoQ and SkQ"

_antioxidants, 2021, doi:10.3390/antiox10040573_

Round 1

Reviewer 1 Report

The authors give a thorough review the pre-clinical and clinical studies which focus on the use of mitochondrial targeted antioxidants (MitoQ, reseveratrol, SKQ1 and SKQR1). The perspective is well written and organized in a meaningful manner. It might be helpful is allowed to include a figure to attract reader attention and understanding of how the specific antioxidants function. One interesting point is also that MitoQ requires function of the ETC--something which might be reduced in diseased states. Would this issue reduce the efficacy of MitoQ in the clinic? 

Author Response

Thank you for the review.  While we did not include a figure, which we felt would have been cumbersome given the widespread effects of treatment, we did assemble a table which summarizes the findings of the pre-clinical studies.  We agree that it is worth speculating on whether damaged mitochondria may be less effective in processing mitochondrial-targeted antioxidants and have included a statement to that effect in the conclusions.  

Reviewer 2 Report

The authors compiled literature on the subject of the use of antioxidants to tackle neurodegenerative diseases. The subject is current despite some controversy and lack of convincing results in human clinical trials. The manuscript is well structured and well written. However, particularly for MitoQ and SkQ1, the sections for clinical studies could be further explored both in terms of context and perspective. Particularly, what is the pre-clinical evidence that led to SkQ1 being tested for macular degeneration and potential mechanisms involved.

What are the authors' views on the underlying justifications for the discrepant observations between pre-clinical and clinical studies? Would antioxidant therapy be far more consistent as co-adjuvant therapy to agents directly targeting primary drivers of disease such as aberrant protein aggregation and accumulation? It is unequivocal that mitochondrial dysfunction and consequent increased production of free radicals and other reactive oxygen species are amongst the earlier events in several neurodegenerative processes, but as the authors discuss in their abstract, it may be the case that these are rather contributors, but not the main pathophysiological events. So could antioxidants afford better outcomes as prophylactics then revealing efficacy at mild or moderate stages of disease development?

Author Response

Thank you for your review.  We have added references to the SkQ1 pre-clinical studies that formed the basis of the current SkQ1 clinical trials in retinal degeneration and multiple sclerosis.  We have also added to the Conclusions in order to speculate on reasons for the lack of efficacy in clinical trials compared to pre-clinical data and note that antioxidants might have potential as a prophylactic.  

Reviewer 3 Report

The authors have nicely summarised the pre- and clinical studies of using resveratrol, mitoQ ans skQ as therapeutics for neurodegenerative diseases.

I would add a table to summarise the findings and to compare the treatments betweek each other.

Author Response

Thank you for your review.  We have added a table summarizing the pre-clinical data.

Round 2

Reviewer 2 Report

The authors satisfactorily addressed the reviewer's comments.